# Surprise Minimizing Multi-Agent Learning with Energy-based Models

**Karush Suri[1], Xiao Qi Shi[2], Konstantinos Plataniotis[1], Yuri Lawryshyn[1]**
[1] University of Toronto, [2] RBC Capital Markets
`karush.suri@mail.utoronto.ca`

## Abstract

Multi-Agent Reinforcement Learning (MARL) has demonstrated significant success by virtue of collaboration across agents. Recent work, on the other hand, introduces *surprise* which quantifies the degree of change in an agent's environment. Surprise-based learning has received significant attention in the case of single-agent entropic settings but remains an open problem for fast-paced dynamics in multi-agent scenarios. A potential alternative to address surprise may be realized through the lens of free-energy minimization. We explore surprise minimization in multi-agent learning by utilizing the free energy across all agents in a multi-agent system. A temporal Energy-Based Model (EBM) represents an estimate of surprise which is minimized over the joint agent distribution. Our formulation of the EBM is theoretically akin to the minimum conjugate entropy objective and highlights suitable convergence towards minimum surprising states. We further validate our theoretical claims in an empirical study of multi-agent tasks demanding collaboration in the presence of fast-paced dynamics. Our implementation and agent videos are available at the `Project Webpage`.

## 1 Introduction

The rise of RL has led to an increasing interest in the study of multi-agent systems [34, 58], commonly known as Multi-Agent Reinforcement Learning (MARL). In the case of partially observable settings, MARL enables the learning of policies with centralised training and decentralised control [26]. This has proven to be useful for exploiting value-based methods which motivate collaboration across large number of agents. *But how do agents behave in the presence of sudden environmental changes?*

Consider the problem of autonomous driving wherein a *driver* (agent) autonomously operates a vehicle in real-time. The *driver* learns to optimize the reward function by maintaining constant speed and covering more distance in different traffic conditions. Whenever the vehicle approaches an obstacle, the *driver* acts to avoid it by utilizing the brake and directional steering commands. However, due to the fast-paced dynamics of the environment, say fast-moving traffic, the agent may abruptly encounter an obstacle (*a person running across the street*) which may result in a collision. Irrespective of the optimal action (*pushing of brakes*) executed by the agent, the vehicle may fail to evade the collision as a result of the abrupt temporal change.

The above arises as a consequence of *surprise*, which is defined as a statistical measure of uncertainty. Surprise minimization [3] is a recent phenomenon observed in the case of single-agent RL methods which deals with environments consisting of rapidly changing states. In the case of model-based RL [24], surprise minimization is used as an effective planning tool in the agent's model [3]. In the case of model-free RL, surprise minimization is witnessed as an intrinsic motivation [1, 36] or generalization problem [9]. On the other hand, MARL does not account for surprise across agents as a result of which agents remain unaware of drastic changes in the environment [35]. Thus, surprise minimization in the case of multi-agent settings requires attention from a critical standpoint.

36th Conference on Neural Information Processing Systems (NeurIPS 2022).

A potential pathway to treat surprising states may be realized in light of free-energy minimization. The free-energy principle depicts convergence to local niches and provides a general recipe for stability among agents. Through this lens, we unify surprise with free-energy in the multi-agent setting. We construct a temporal EBM which represents an estimate of surprise agents may face in the environment. All agents jointly minimize this estimate utilizing temporal difference learning upon their value functions and the EBM. Our formulation of free-energy minimization is theoretically akin to minimizing the entropy in conjugate gradient space. This insight provides a suitable convergence result towards minimum surprising states (or niches) of the agent state distributions. In an empirical study of multi-agent tasks which present significant collaboration bottlenecks and fast-paced dynamics, we validate our theoretical claims and motivate the practical usage of EBMs in MARL.

## 2 Related Work

**Surprise Minimization:** Despite the recent success of value-based methods [39, 22] RL agents suffer from spurious state spaces and encounter sudden changes in trajectories. Quantitatively, surprise has been studied as a measure of deviation [3, 9] among states encountered by the agent during its interaction with the environment. While exploring [7, 56] the environment, agents tend to have higher deviation among states which is gradually reduced by gaining a significant understanding of state-action transitions. In the case of model-based RL, agents can leverage spurious experiences [3] and plan effectively for future steps. On the other hand, in the case of model-free RL, surprise results in sample-inefficient learning [1]. This is primarily addressed by making use of rigorous exploration strategies [52, 31]. High-dimensional exploration further requires extrinsic feature engineering [27] and meta models [16]. A suitable way to tackle high-dimensional dynamics is by utilizing surprise as a penalty on the reward [9]. This leads to improved generalization for single-agent interactions [45]. Our proposed approach is parallel to the aforesaid methods.

**Energy-based Models:** EBMs have been successfully implemented in single-agent RL methods [42, 19]. These typically make use of Boltzmann distributions to approximate policies [32]. Such a formulation results in the minimization of free energy within the agent. While policy approximation depicts promise in the case of unknown dynamics, inference methods [57] play a key role in optimizing goal-oriented behavior.

A second type of usage of EBMs follows the maximization of entropy [65]. The maximum entropy framework [20] highlighted in Soft Q-Learning (SQL) [19] allows the agent to obey a policy which maximizes its reward and entropy concurrently. Maximization of agent's entropy results in diverse and adaptive behaviors [64] which may be difficult to accomplish using standard exploration techniques [7, 56]. The maximum entropy framework is akin to approximate inference in the case of policy gradient methods [49]. Such a connection between likelihood ratio gradient techniques and energy-based formulations leads to diverse and robust policies [17]. Furthermore, their hierarchical extensions [18] preserve the lower levels of hierarchies. In the case of MARL, EBMs have witnessed limited applicability as a result of the increasing number of agents and complexity within each agent [8]. While the framework is readily transferable to opponent-aware multi-agent systems [63], cooperative settings consisting of coordination between agents require a firm formulation of energy. This formulation must be scalable in the number of agents [15] and account for environments consisting of spurious states [62]. Our theoretical formulation is motivated by these methods in literature.

## 3 Preliminaries

**Multi-Agent Learning:** We review the cooperative MARL setup. The problem is modeled as a Decentralized Partially Observable Markov Decision Process (Dec-POMDP) [43] defined by the tuple $(\mathcal{S}, \mathcal{A}, r, N, P, Z, O, \gamma)$ where the state space $\mathcal{S}$ and action space $\mathcal{A}$ are discrete, $r : \mathcal{S} \times \mathcal{A} \rightarrow [r_{min}, r_{max}]$ presents the reward observed by agents $a \in N$ where $N$ is the set of all agents, $P : \mathcal{S} \times \mathcal{S} \times \mathcal{A} \rightarrow [0, 1]$ presents the unknown transition model consisting of the transition probability to the next state $s' \in \mathcal{S}$ given the current state $s \in \mathcal{S}$ and joint action $u \in \mathcal{A}$ (a combination of each agent's action $u^a \in \mathcal{A}^a$) at time step $t$ and $\gamma$ is the discount factor. Our setting consists of the finite-horizon discounted problem case where episodes terminate at timestep $T$ with the terminal state being $s_T$. As a result, task returns remain bounded for each episode. We consider a partially observable setting in which each agent $n$ draws individual observations $z \in Z$ according to the observation function $O(s, u) : \mathcal{S} \times \mathcal{A} \rightarrow Z$. We consider a joint policy $\pi_\theta(u|s)$ as a function of

model parameters $\theta$. Standard RL defines the agent's objective to maximize the expected discounted reward $\mathbb{E}_{\pi_\theta}[\sum_{t=0}^T \gamma^t r(s_t, u_t)]$ as a function of the parameters $\theta$. The joint action-value function for agents is represented as $Q(u, s; \theta) = \mathbb{E}_{\pi_\theta}[\sum_{t=1}^T \gamma^t r(s, u)|s = s_t, u = u_t]$ which is the expected sum of payoffs obtained in state $s$ upon performing action $u$ by following the policy $\pi_\theta$. We denote the optimal policy $\pi_{\theta*}$ (shorthand $\pi^*$) such that $Q(u, s; \theta^*) \geq Q(u, s; \theta) \forall s \in S, u \in A$. In the case of multiple agents, the joint optimal policy can be expressed as the Nash Equilibrium [40] of the Stochastic Markov Game as $\pi^* = (\pi^{1,*}, \pi^{2,*}, ... \pi^{N,*})$ such that $Q(u^a, s; \theta^*) \geq Q(u^a, s; \theta) \forall s \in S, u \in A, a \in N$. Q-Learning is an off-policy, model-free algorithm suitable for continuous and episodic tasks. The algorithm uses semi-gradient descent to minimize the Temporal Difference (TD) error in Equation 1.

$$\mathbb{L}(\theta) = \mathbb{E}_{s,u,s'\sim\mathcal{R}}\left[\left(r + \gamma\max_{u'\in A}Q(u', s'; \theta^-) - Q(u, s; \theta)\right)^2\right] \tag{1}$$

where $y = r + \gamma\max_{u'\in A}Q(u', s'; \theta^-)$ is the TD target consisting of $\theta^-$ as the target parameters and $\mathcal{R}$ denotes the replay buffer.

**Energy-based Models:** EBMs [29, 30] have been successfully applied in the field of machine learning [55] and probabilistic inference [37]. A typical EBM $\mathcal{E}$ formulates the equilibrium probabilities [47] $P(v, h) = \frac{\exp(-\mathcal{E}(v,h))}{\sum_{\hat{v},\hat{h}}[\exp(-\mathcal{E}(\hat{v},\hat{h}))]}$ via a Boltzmann distribution [32] where $v$ and $h$ are the values of the visible and hidden variables and $\hat{v}$ and $\hat{h}$ are all the possible configurations of the visible and hidden variables respectively. The probability distribution over all the visible variables can be obtained by summing over all possible configurations of the hidden variables. This is mathematically expressed in Equation 2.

$$P(v) = \frac{\sum_h \exp(-\mathcal{E}(v, h))}{\sum_{\hat{v},\hat{h}} \exp(-\mathcal{E}(\hat{v}, \hat{h}))} \tag{2}$$

Here, $\mathcal{E}(v, h)$ is called the equilibrium free energy which is the minimum of the variational free energy and $\sum_{\hat{v},\hat{h}} \exp(-\mathcal{E}(\hat{v}, \hat{h}))$ is the partition function.

## 4 Energy-based Surprise Minimization

We begin by constructing surprise minimization as an energy-based problem in the temporal setting. The motivation behind an energy-based formulation stems from rapidly changing states as an undesired niche among agents in the case of partially-observed settings. To steer agents away from this niche, we further construct a method which incorporates the theoretical aspect of the study.

### 4.1 The Surprise Minimization Objective

To make analysis tractable towards valid function spaces and surprising states, we take into account two assumptions which form the central basis of surprise minimization among multiple agents.

> **Assumption 1.** (Completeness of value function space) *The space $\Pi : \mathcal{S} \times \mathcal{A}$ of all Q value functions $Q(s, u) \in \Pi$, $\forall s \in \mathcal{S}$, $\forall u \in \mathcal{A}$ is a nonempty complete metric space.*

Assumption 1 restricts the formulation of individual agent value functions $Q_a$ to the nonempty complete metric space. A nonempty space confirms the presence of candidate functions $Q_a$ upper bounded by the optimal function $Q^*$, i.e.- $Q_a \leq Q^*$, $\forall a \in N$ [5]. The completeness counterpart, on the other hand, provisions a fixed interior **int** $\Pi$ for optimization [6].

> **Assumption 2.** (Constant surprise at Equilibrium) *In the limit of convergence $\lim_{\pi_a \to \pi^*}$ to an optimal policy $\pi^*$, all agents $a \in N$ incur a finite surprise $\zeta > 0$ between consecutive states $s$ and $s'$ until termination state $s_T$.*

Assumption 2 is directly based on the constant and continuous temporal aspect of surprise minimization [50, 12]. Corresponding to the lifetime of each agent $a \in N$, a desired minima bakes in the optimal distribution of actions which correspond to minimum but finite instantaneous surprise.

We formulate the energy-based objective consisting of surprise as a function of states $s$, joint actions $u$ and standard deviation $\sigma$ of observations for each agent $a$. In the case of high-dimensional state spaces (such as multiple opponents), $\sigma$ informs agents of the abrupt statistical change that would take place upon executing action $u$. We formulate surprise as $\mathcal{T}V_{\text{surp}}^a(s, u, \sigma)$ which serves as an uncertainty quantifier Unc(s,a) of the state-action distribution. Here $V_{\text{surp}}^a(s, u, \sigma)$ denotes the surprise value function which serves as a mapping from agent and environment dynamics to surprise. Define an operator presented in Equation 3 which sums surprising configurations across all agents.

$$\mathcal{T}V_{\text{surp}}^a(s, u, \sigma) = \log \sum_{a=1}^{N} \exp\left(V_{\text{surp}}^a(s, u, \sigma)\right) \tag{3}$$

**Remark 1.** *$\mathcal{T}V_{\text{surp}}^a(s, u, \sigma)$ intuitively provides a global estimate of surprise. If all agents are equally likely to face a surprising state, then $\mathcal{T}V_{\text{surp}}^a(s, u, \sigma)$ captures their individual contributions.*

The formulation makes use of the soft-maximum operator [2]. The operator $\mathcal{T}V_{\text{surp}}^a(s, u, \sigma)$ is similar to prior energy formulations [19] where the energy across different actions is evaluated. In our case, inference is carried out across all agents with actions as prior variables. However, in the special case of using an EBM as a $Q$-function, our approach suitably generalizes to the above methods (details in Appendix B).

Our choice of $\mathcal{T}V_{\text{surp}}^a(s, u, \sigma)$ is based on its unique mathematical properties which result in better convergence. Of these properties, the most useful result is that $\mathcal{T}$ forms a contraction on the surprise value function $V_{\text{surp}}^a(s, u, \sigma)$ indicating a guaranteed minimization of surprise within agents. This is formally stated in Theorem 1 while utilizing the completeness criterion of Assumption 1 which provides a tractable value function space. All proofs are deferred to Appendix A.

> **Theorem 1.** *Given a surprise value function $V_{\text{surp}}^a(s, u, \sigma) \ \forall a \in N$, the energy operator $\mathcal{T}V_{\text{surp}}^a(s, u, \sigma) = \log \sum_{a=1}^{N} \exp\left(V_{\text{surp}}^a(s, u, \sigma)\right)$ forms a contraction on $V_{\text{surp}}^a(s, u, \sigma)$.*

Theorem 1 provides a suitable guarantee of $\mathcal{T}V_{\text{surp}}^a(s, u, \sigma)$ converging to a fixed point niche. The contraction result is directly based on Banach's fixed point property and suggests the generalization of convergence in any nonempty complete metric space $(X, d)$ [5].

We now consider a weighted combination of $Q(s, u)$ with $\mathcal{T}V_{\text{surp}}^a(s, u, \sigma)$ wherein we denote $\beta$ as a temperature parameter,

$$\hat{Q}(u, s; \theta) = Q(u, s; \theta) + \beta \log \sum_{a=1}^{N} \exp\left(V_{\text{surp}}^a(s, u, \sigma)\right)) \tag{4}$$

**Remark 2.** *Equation 4 is an instance of value function regularization wherein the Q values are subject to a joint penalty while observing surprising states.*

Interestingly, upon considering the Legendre transform $f^*(x)$ [6, 14] (convex conjugate function corresponding to the conjugate space $\mathcal{X}$ of a differentiable function $f(z)$) of $\mathcal{T}V_{\text{surp}}^a(s, u, \sigma)$, we obtain the following,

$$f^*(x) = \sup_{z \in \text{dom} f} \left(x^{\text{T}}z - f(z)\right) \ , \ f(z) = \mathcal{T}V_{\text{surp}}^a(s, u, \sigma) \tag{5}$$

$$f^*(x) = \sum_{x} x \log(x) \ , \ x = \nabla_z f(z) \in \mathcal{X} \tag{6}$$

**Remark 3.** *The Legendre Transform of $\mathcal{T}V_{\text{surp}}^a(s, u, \sigma)$ given by $f^*(x) = \sum_{x} x \log(x)$ when utilized as value function regularization $\hat{Q} = Q - f^*(x)$ corresponds to the minimum entropy formulation in conjugate space $\mathbb{E}_{\pi_\theta}\left[\sum_{t=0}^{T} \gamma^t(r(s_t, u_t) - \lambda \mathcal{H}(x))\right]$ for $x = \nabla_z f(z) \in \mathcal{X}$.*

Based on the above insight, minimizing entropy to express $\nabla_z f(z)$ in conjugate space is akin to minimizing uncertainty among all agents in the value function space $\Pi$. Intuitively, $\mathcal{H}(x)$ denotes the uncertainty for each agent $a \in N$ in the multi-agent population which is directly related to its

ability of accurately interpreting the environment. Minimizing $\mathcal{H}(x)$ leads to an increase in the expressiveness of value function. This in turn, induces an expressive state visitation distribution which steers the agent away from sudden changes in its environment. Note that the setting does not minimize entropy in value function space which would stand contrary to the maximum entropy formulation [20] (see Appendix B).

Figure 1 presents an illustration of the intuition behind surprise minimization using the energy-based scheme. Agents collaborate in partially-observed worlds to attain a joint niche. Interpreting the space of all surprising states as an energy landscape, MARL agents move from high energy states to low energy states which consist of minimum surprise. During training, agents train to find policies which not only provide rewarding actions, but also avoid risky states by minimizing $\mathcal{T}V_{\text{surp}}^a(s, u, \sigma)$. Seeking these states leads to finding the minima on the energy landscape. Thus, it is by virtue of regularized value estimates $\hat{Q}$ that the minimization scheme informs agents of joint surprise.

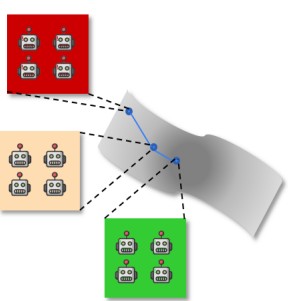

Figure 1: Agent populations (robots) traverse the energy landscape (in grey) during update steps (blue) to seek energy minima (darker shade at center). This results in surprise minimization from high (red) to low energy (green) niches.

## 4.2 Surprise Minimization with Function Approximation

We utilize the above insights as surprise-based regularization in the TD learning setting. Upon replacing $Q(u, s; \theta)$ with $\hat{Q}(u, s, ; \theta)$ in the RL construction of Equation 1 one obtains the following,

$$L(\theta) = \mathbb{E}_{s,u,s' \sim \mathcal{R}} \left[ \frac{1}{2} \left( \hat{y} - (Q(u, s; \theta) + \beta \log \sum_{a=1}^{N} \exp\left(V_{\text{surp}}^a(s, u, \sigma)\right)) \right)^2 \right]$$

where $\hat{y}$ is given by the following expression,

$$\hat{y} = r + \gamma \max_{u'} Q(u', s'; \theta^-) + \beta \log \sum_{a=1}^{N} \exp\left(V_{\text{surp}}^a(s', u', \sigma')\right) \tag{7}$$

Collecting the log terms yields the following,

$$L(\theta) = \mathbb{E}_{s,u,s' \sim \mathcal{R}} \left[ \frac{1}{2} \left( r + \gamma \max_{u'} Q(u', s'; \theta^-) \right. \right.$$
$$\left. \left. + \beta \log \left( \frac{\sum_{a=1}^{N} \exp\left(V_{\text{surp}}^a(s', u', \sigma')\right)}{\sum_{a=1}^{N} \exp\left(V_{\text{surp}}^a(s, u, \sigma)\right)} \right) - Q(u, s; \theta) \right)^2 \right]$$

$$L(\theta) = \mathbb{E}_{s,u,s' \sim \mathcal{R}} \left[ \frac{1}{2} \left( r + \gamma \max_{u'} Q(u', s'; \theta^-) + \beta E - Q(u, s; \theta) \right)^2 \right] \tag{8}$$

Here, $E$ is defined as the *surprise ratio*. The surprise value function $V_{\text{surp}}^a(s', u', \sigma')$ is expressed as the negative free energy and $\sum_{a=1}^{N} \exp\left(V_{\text{surp}}^a(s, u, \sigma)\right)$ as the partition function of a conventional EBM described in Equation 2. Alternatively, $V_{\text{surp}}^a(s, u, \sigma)$ can be formulated as the negative free energy with $\sum_{a=1}^{N} \exp\left(V_{\text{surp}}^a(s', u', \sigma')\right)$ as the partition function. The TD objective incorporates the minimization of surprise across all agents as minimizing the energy in rapidly changing states.

**Remark 4.** *The above formulation of $\beta E$ can be realized as intrinsic motivation steering the agent towards subgoals with reduced surprise.*

The energy formulation $E$ provides a tractable distribution over all surprising configurations in the state space $\mathcal{S}$. This guarantees convergence to minimum surprise at optimal policy $\pi^*$ and is formally expressed in Theorem 2 (see Appendix C for a detailed convergence analysis).

**Theorem 2.** *Upon agent's convergence to an optimal policy $\pi^*$, total energy of $\pi^*$, expressed by $E^*$ will reach a thermal equilibrium consisting of minimum surprise among consecutive states $s$ and $s'$.*

Theorem 2 demonstrates an intuitive convergence result of agent populations collaborating to reside in a mutual ecological niche [12]. The multi-agent population with minimum surprise exhibits the optimal policy $\pi^*$ which results in minimum energy corresponding to each surprising state in the state distribution $\mathcal{S}$. Orthogonally, agents may continue to experience finite and constant surprise in the long-horizon while acting optimally to visit non-surprising and rewarding states. This presents surprise minimization as a secondary surrogate objective in MARL.

### 4.3 Energy-based MIXer (EMIX)

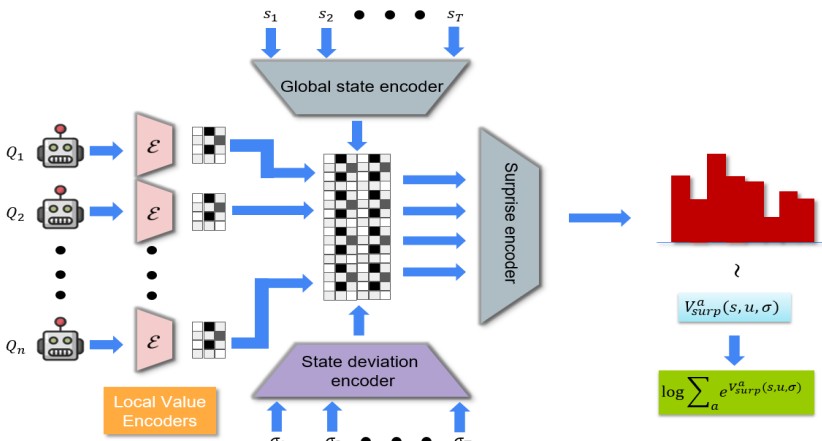

Figure 2: The EMIX architecture for learning surprise across global states.

Based on our theoretical analysis, we incorporate learning of surprise as global intrinsic motivation across all agents in the multi-agent system. A global estimate of surprise, following the energy operator $\mathcal{T}V_{\text{surp}}^a(s, u, \sigma)$, is befitting from a computational perspective as well. An individual estimate of surprise for each agent may be intractable to obtain due to the non-stationarity of the environment. Instead, we seek to minimize surprise jointly across all agents using an expressive Energy-based MIXer (EMIX) architecture which is compatible with any multi-agent RL algorithm. Figure 2 illustrates our learning scheme.

Learning of surprise in the high-dimensional value function space is cumbersome with the number of actions scaling linearly in the number of agents. This imposes an inherent restriction to learn global surprise efficaciously across all agents at a given timestep. Towards this goal, EMIX encodes individual value functions $Q_1, Q_2, \dots Q_n$ corresponding to each agent using local value encoders. These encoders capture the local change in value functions arising over subsequent TD learning iterations [60]. A global state encoder maps environment states $s_1, s_2, \dots s_T$ to a low dimensional representation. Further, a state deviation encoder encodes deviations across all states $s_1, s_2, \dots s_T$ within the given batch. Akin to a model-based method [23], the state deviation encoder accounts for uncertainty in an agent's state visitation distribution. Note that the encoder does not construct an explicit model of states, but only represents their variation in the agent's environment. This insight is essential to account for abrupt dynamics encountered by agents. Representations obtained from state and value function encoders are concatenated and compressed using a final surprise encoder which estimates a distribution of surprise values. The distribution implicitly represents the density of states wherein an agent may encounter most surprise. A value estimate $V_{\text{surp}}^a(s, u, \sigma)$ sampled from the surprise distribution depicts the variational free energy configuration upon application of $\mathcal{T}$ which serves as global intrinsic motivation. Practical training of EMIX proceeds with backpropagation [46] using gradient descent and the reparameterization trick [25] for sampling of $V_{\text{surp}}^a(s, u, \sigma)$.

## 4.4 Practical Implementation

Algorithm 1 presents the EMIX framework (in green) combined with QMIX [44], an off-the-shelf MARL algorithm. The total $Q$-value $Q_{tot}^\theta$ is computed by the mixer network with its inputs as the $Q$-values of all the agents conditioned on $s$ via the hypernetworks. Similarly, the target mixers approximate $Q_i^{\theta^-}$ conditioned on $s'$. In order to evaluate surprise within agents, we compute the standard deviations $\sigma$ and $\sigma'$ across all observations $z$ and $z'$ for each agent using $s$ and $s'$ respectively. The surprise value function, called the Surprise-Mixer, estimates surprise $V_{\text{surp}}^a(s, u, \sigma)$ conditioned on $s$, $u$ and $\sigma$. The same computation is repeated using the Target-Surprise-Mixer for estimating surprise $V_{\text{surp}}^a(s', u', \sigma')$ within next-states in the batch. Application of the energy operator along the non-singleton agent dimension for $V_{\text{surp}}^a(s, u, \sigma)$ and $V_{\text{surp}}^a(s', u', \sigma')$ yields the energy ratio $E$ which is used in Equation 8 to evaluate $L(\theta)$. We then use batch gradient descent to update parameters of the mixer $\theta$. Target parameters $\theta_i^-$ are updated every $update-interval$ steps.

---

**Algorithm 1** Energy-based MIXer (EMIX)

---

1: Initialize $\phi, \theta, \theta_1^- ..., \theta_m^-$, agent and hypernetwork parameters.
2: Initialize learning rate $\alpha$, temperature $\beta$ and replay buffer $\mathcal{R}$.
3: **for** environment step **do**
4:     $u \leftarrow (u_1, u_2..., u_N)$
5:     $\mathcal{R} \leftarrow \mathcal{R} \cup \{(s, u, r, s')\}$
6:     **if** $|\mathcal{R}| >$ batch-size **then**
7:         **for** random batch **do**
8:             $Q_{tot}^\theta \leftarrow$ *Mixer-Network*$(Q_1, Q_2..., Q_N, s)$
9:             $Q_i^{\theta^-} \leftarrow$ *Target-Mixer$_i$*$(Q_1, Q_2..., Q_N, s'), \forall i = 1, 2.., m$
10:           **Calculate $\sigma$ and $\sigma'$ using $s$ and $s'$**
11:           $V_{\text{surp}}^a(s, u, \sigma) \leftarrow$ ***Surprise-Mixer**(s, u, \sigma)*
12:           $V_{\text{surp}}^a(s', u', \sigma') \leftarrow$ ***Target-Mixer**(s', u', \sigma')*
13:           $E \leftarrow \log \left( \frac{\sum_{a=1}^N \exp\left(V_{\text{surp}}^a(s', u', \sigma')\right)}{\sum_{a=1}^N \exp\left(V_{\text{surp}}^a(s, u, \sigma)\right)} \right)$
14:           **Calculate $L(\theta)$ using $E$ in Equation 8**
15:           $\theta \leftarrow \theta - \alpha \nabla_\theta L(\theta)$
16:         **end for**
17:     **end if**
18:     **if** update-interval steps have passed **then**
19:         $\theta_i^- \leftarrow \theta, \forall i = 1, 2.., m$
20:     **end if**
21: **end for**

---

## 5 Experiments

Our experiments aim to evaluate the theoretical claims presented by EMIX along with its performance to prior MARL methods. Specifically, we aim to answer the following questions;

**(1)** How does the provision of an EBM for surprise minimization compare to current MARL methods?

**(2)** Does the algorithm validate the theoretical claims corresponding to its components?

### 5.1 Energy-based Surprise Minimization

We assess the validity of EMIX, when combined with QMIX, on multi-agent StarCraft II microman-agement scenarios [48] as these consist of a larger number of agents with different action spaces. This in turn motivates a greater deal of coordination. Additionally, micromanagement scenarios in StarCraft II consist of multiple opponents which introduce a greater degree of surprise within consecutive states.

We compare our method to prior methods namely; (1) QMIX [44], constituting of nonlinear value function factorization with monotonicity constraints; (2) Value Decomposition Networks (VDN)

| Scenarios | EMIX | SMiRL-QMIX | QMIX | VDN | COMA | IQL |
|---|---|---|---|---|---|---|
| 3m | **94.90±0.39** | 93.94±0.22 | 93.43±0.20 | 94.58±0.58 | 84.75±7.93 | 94.79±0.50 |
| 3s_vs_4z | **97.22±0.73** | 0.24±0.11 | 96.01±3.93 | 94.29±2.13 | 0.00±0.00 | 59.75±12.22 |
| 8m_vs_9m | **71.03±2.69** | 69.90±1.94 | **68.28±2.30** | 58.81±4.68 | 4.17±0.58 | 28.48±22.38 |
| 10m_vs_11m | 75.35±2.30 | **77.85±2.02** | 70.36±2.87 | 71.81±6.50 | 4.55±0.73 | 32.27±25.68 |
| so_many_baneling | **95.87±0.16** | 93.61±0.94 | 93.35±0.78 | 92.26±1.06 | 91.65±2.26 | 74.97±6.52 |
| 5m_vs_6m | **37.07±2.42** | 33.27±2.79 | 34.42±2.63 | 35.63±3.32 | 0.52±0.13 | 14.78±2.72 |

Table 1: Comparison of success rate percentages between EMIX and prior MARL methods on StarCraft II micromanagement scenarios. EMIX is comparable to or improves over QMIX agent. In comparison to SMiRL-QMIX, EMIX demonstrates improved minimization of surprise. Results are averaged over 5 random seeds.

[53], consisting of linear additive factorization of $Q$ function; (3) Counterfactual Multi-Agent Policy Gradients (COMA) [11], which consist of counterfactual actor-critic updates in a centralized critic; and (4) Independent $Q$ Learning (IQL) [54], wherein each agent acts independent of other agents. (5) In order to compare our surprise minimization scheme against pre-existing mechanisms, we compare EMIX additionally to a model-free implementation of SMiRL [3] in QMIX. We use the generalized version of SMiRL as it demonstrates reduced variance across batches [9]. This implementation is denoted as SMiRL-QMIX for comparisons. Details related to the implementation of EMIX are presented in Appendix D.

Table 5 presents the comparison of success rate percentages between EMIX and prior MARL algorithms on 6 StarCraft II micromanagement scenarios. Corresponding to each scenario, algorithms demonstrating higher success rate values in comparison to other methods have their entries highlighted in **bold** (see Appendix E.1 for a statistical analysis). Out of the 6 scenarios considered, EMIX presents higher success rates on 5 of these scenarios depicting the suitability of the proposed approach. In cases of *so_many_baneling* and *5m_vs _6m* having large number of opponents and a greater level of surprise, EMIX aptly improves over prior methods. When compared to QMIX, EMIX depicts improved success rates on all of the 6 scenarios. On comparing EMIX with SMiRL-QMIX, EMIX demonstrates higher average success rates indicating surprise robust policies.

## 5.2 Ablation Study

We now present the ablation study for the various components of EMIX. Our experiments aim to determine the effectiveness of the energy-based surprise minimization method. Additionally, we also aim to evaluate the utility of dual approximators for surprise estimation in accordance with the precept from RL literature [21, 13, 20].

**EMIX Objective:** To weigh the effectiveness of energy-based scheme, we ablate the energy operator $\mathcal{T}$ and only utilize $V_{\mathrm{surp}}^a$. Since this implementation employs dual approximators $V_{\mathrm{surp},(i)}^a$ $i \in \{1, 2\}$ for stability, we call this implementation as TwinQMIX. Thus, we compare between QMIX, TwinQMIX and EMIX to assess the contributions of each of the proposed methods.

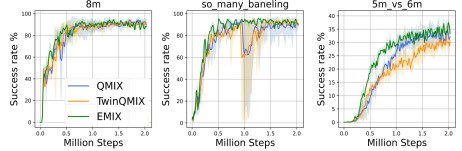

Figure 3: Ablations for each of EMIX's component. When compared to QMIX, EMIX and TwinQMIX depict improvements in performance and sample efficiency.

Figure 3 presents the comparison of average success rates for QMIX, TwinQMIX and EMIX on 3 different scenarios. In comparison to QMIX, TwinQMIX adds stability to the original objective by incorporating surprising estimates. On comparing TwinQMIX to EMIX we note that dual approximators play little role in improving convergence. Thus, the energy-based surprise minimization scheme is the main facet for significant performance improvement. This is demonstrated in the *5m_vs_6m* scenario wherein the EMIX implementation improves the performance of TwinQMIX in comparison to QMIX by utilizing a surprise-robust policy. In the case of *so_many _baneling* scenario which consists of a large number of opponents (27 banelings), EMIX tackles surprise effectively by preventing a significant drop in performance which is observed in cases of QMIX and TwinQMIX.

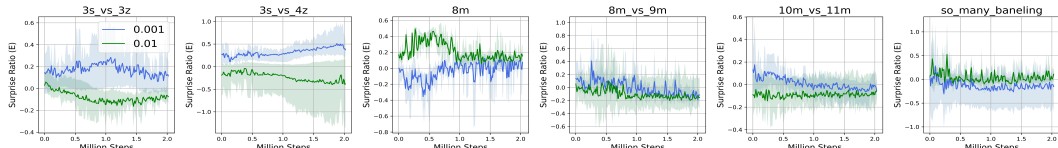

Figure 4: Variation of surprise minimization with temperature $\beta$. Learning of surprise is achieved by making use of a suitable value of temperature parameter ($\beta = 0.01$) which controls the stability in surprise minimization by utilizing $E$ as intrinsic motivation.

**Surprise Minimization with Temperature:** The importance of $\beta$ can be validated by assessing its usage in surprise minimization. We observe the variation of $E$ as it is a collection of surprise-based sample estimates across the batch. Additionally, $E$ consists of prior samples $V_{\mathrm{surp}}^a(s, u, \sigma)$ for $V_{\mathrm{surp}}^a(s', u', \sigma')$ which makes inference tractable.

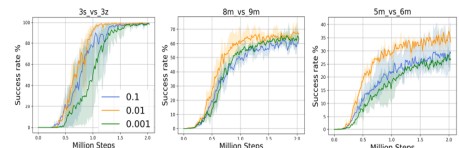

Figure 5: Variation in success rates with temperature $\beta$. A value of $\beta = 0.01$ is found to work best.

Figure 4 presents the variation of Energy ratio $E$ with the temperature parameter $\beta$ during learning. We compare two stable variations of $E$ at $\beta = 0.001$ and $\beta = 0.01$. The objective minimizes $E$ over the course of learning and attains thermal equilibrium with minimum energy. Intuitively, equilibrium corresponds to convergence to optimal policy $\pi^*$ which validates the claim in Theorem 2. With $\beta = 0.01$, EMIX presents improved convergence and surprise minimization for 5 out of the 6 considered scenarios, hence validating the suitable choice of $\beta$. The choice of $\beta$ is further validated in Figure 5 wherein $\beta = 0.01$ provides consistent stable improvements over other values. Lower values of $\beta$, such as $\beta = 0.001$, do little to minimize surprise or improve performance.

# 6 Qualitative Analysis

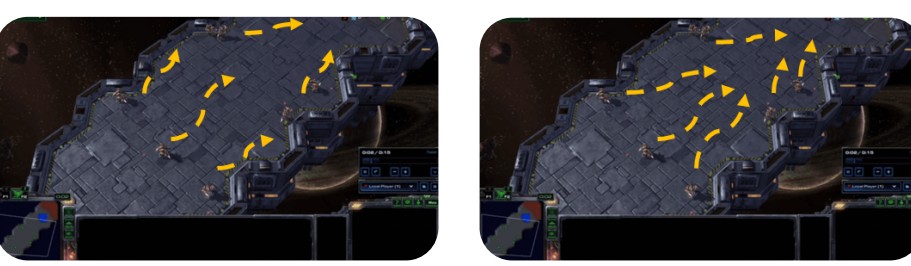

Figure 6: Task- *so_many_baneling*, **(left)** Behaviors learned by EMIX agents, **(right)** Behaviors learned by QMIX agents

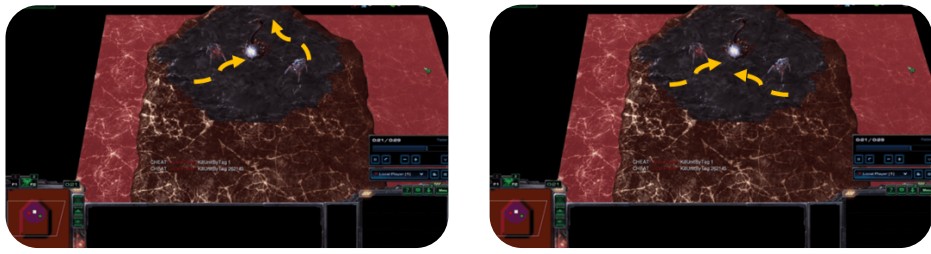

Figure 7: Task- *2s_vs_1sc*, **(left)** Behaviors learned by EMIX agents, **(right)** Behaviors learned by QMIX agents

We visualize and compare behaviors learned by surprise minimizing agents to the prior method of QMIX. Fig. 6 presents the comparison of EMIX and QMIX agent trajectories (in yellow arrows) on

the challenging *so_many_baneling* task. The task consists of 27 baneling opponents which rapidly attack the agent team on a bridge. QMIX agents naively move to the central alley of the bridge and start attacking enemies early on. While QMIX agents naively maximize returns, EMIX agents learn a different strategy. EMIX agents rearrange themselves first at the corners of the bridge. Note that these corners provide cover from enemy's fire. Thus, EMIX agents learn to take cover before approaching the enemy head-on. This indicates that the surprise-robust policy is aware of the incoming fast-paced assault.

As another example, Fig. 7 presents behaviors on the *2s_vs_1sc* task wherein two agents must collaborate together to defeat a SpineCrawler enemy. The enemy, having a long tentacle pointing to the front, chooses to attack any one of the agents **randomly** in front of it. Additionally, the tentacle has a fixed length and cannot extend beyond this range. Random intermittent attacks indicate that the agents face a greater degree of surprise with no prior knowledge of the enemy's movement. We observe that QMIX agents take turns to attack the enemy by moving back and forth to minimize damage. EMIX agents, on the other hand, learn a different strategy. One of the EMIX agents stands at a distance to attack th enemy while the other agent goes around to attack from behind. This indicates that the policy is aware of enemy's limited movement.

## 6.1 Predator-Prey Benchmark

We extend our comparison of EMIX on the Predator-Prey (particle world) tasks. In addition to the difficulty of task, we vary the number of opponents. This helps quantify the variation in performance against increasing level of surprise under fixed dynamics. Table 2 presents average returns. While all agents present comparable performance on the easier tasks, EMIX improves over QMIX and TwinQMIX on the more challenging *punish* and *hard* tasks. In the case of *punish*, EMIX is the only method to achieve greater than 20 returns. Additional results can be found in Appendix E.3.

| Scenarios | EMIX | TwinQMIX | SMiRL-QMIX | QMIX | VDN | COMA | IQL |
|---|---|---|---|---|---|---|---|
| predator_prey_easy | **40.00 ± 0.13** | 40.00 ± 0.34 | 40.00 ± 0.98 | 40.00 ± 0.22 | 38.74 ± 0.64 | 27.49 ± 4.26 | 34.73 ± 2.92 |
| predator_prey | 40.00 ± 0.72 | 40.00 ± 1.92 | 40.00 ± 0.27 | **40.00 ± 0.16** | 36.23 ± 3.19 | 25.13 ± 0.92 | 31.59 ± 0.74 |
| predator_prey_punish | **24.17 ± 3.29** | 20.32 ± 4.15 | 19.31 ± 1.12 | 14.33 ± 3.81 | 17.21 ± 2.31 | 10.92 ± 4.35 | 7.86 ± 3.21 |
| predator_prey_hard | **12.34 ± 3.11** | 10.19 ± 1.15 | 10.47 ± 0.83 | 8.76 ± 4.33 | 5.19 ± 3.97 | -4.37 ± 1.53 | -9.26 ± 4.84 |

Table 2: Comparison of average returns between EMIX, its ablations and prior MARL methods on Predator-Prey tasks. EMIX improves over QMIX and SMiRL-QMIX.

## 7 Discussion

**Conclusion:** In this paper, we presented an energy-based perspective towards surprise minimization in multi-agent RL. Towards this goal we introduce EMIX, an energy-based intrinsic motivation framework for surprise minimization in MARL algorithms. EMIX utilizes a temporal EBM to estimate and minimize surprise jointly across all agents. Our theoretical claims on the formulation of minimization of temporal energy with surprise are corroborated upon utilizing EMIX on a suite of challenging MARL tasks requiring significant collaboration under fast-paced dynamics.

**Future Work:** While EMIX serves as a practical example of EBMs in cooperative MARL, it presents several new avenues for future work. We shed light on 2 such aspects,

(1) Provision of an energy-based model naturally raises the question of *how can we efficiently sample from the surprise distribution?* Advances in sampling methods depict promise towards this aspect.

(2) Although suitable for lower dimensions, the scalability of EBMs towards high dimensional action spaces remains an open question. We conjecture that the utility of density-based methods and generative models can address the scalability gap. These directions are left for future work.

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
