# A Proofs

**Theorem 1.** *Given a surprise value function $V_{\text{surp}}^a(s, u, \sigma) \, \forall a \in N$, the energy operator $\mathcal{T}V_{\text{surp}}^a(s, u, \sigma) = \log \sum_{a=1}^{N} \exp\left(V_{\text{surp}}^a(s, u, \sigma)\right)$ forms a contraction on $V_{\text{surp}}^a(s, u, \sigma)$.*

*Proof.* We follow the process of [2]. Let us first define a norm on surprise values $||V_1 - V_2|| \equiv \max_{s,u,\sigma}|V_1(s, u, \sigma) - V_2(s, u, \sigma)|$. Suppose $\epsilon = ||V_1 - V_2||$,

$$\log \sum_{a=1}^{N} \exp\left(V_1(s, u, \sigma)\right) \leq \log \sum_{a=1}^{N} \exp\left(V_2(s, u, \sigma) + \epsilon\right)$$

$$= \log \sum_{a=1}^{N} \exp\left(V_1(s, u, \sigma)\right) \leq \log \exp\left(\epsilon\right) \sum_{a=1}^{N} \exp\left(V_2(s, u, \sigma)\right)$$

$$= \log \sum_{a=1}^{N} \exp\left(V_1(s, u, \sigma)\right) \leq \epsilon + \log \sum_{a=1}^{N} \exp\left(V_2(s, u, \sigma)\right)$$

$$= \log \sum_{a=1}^{N} \exp\left(V_1(s, u, \sigma)\right) - \log \sum_{a=1}^{N} \exp\left(V_2(s, u, \sigma)\right) \leq ||V_1 - V_2|| \tag{9}$$

Similarly, using $\epsilon$ with $\log \sum_{a=1}^{N} \exp\left(V_1(s, u, \sigma)\right)$,

$$\log \sum_{a=1}^{N} \exp\left(V_1(s, u, \sigma) + \epsilon\right) \geq \log \sum_{a=1}^{N} \exp\left(V_2(s, u, \sigma)\right)$$

$$= \log \exp\left(\epsilon\right) \sum_{a=1}^{N} \exp\left(V_1(s, u, \sigma)\right) \geq \log \sum_{a=1}^{N} \exp\left(V_2(s, u, \sigma)\right)$$

$$= \epsilon + \log \sum_{a=1}^{N} \exp\left(V_1(s, u, \sigma)\right) \geq \log \sum_{a=1}^{N} \exp\left(V_2(s, u, \sigma)\right)$$

$$= ||V_1 - V_2|| \geq \log \sum_{a=1}^{N} \exp\left(V_2(s, u, \sigma)\right) - \log \sum_{a=1}^{N} \exp\left(V_1(s, u, \sigma)\right) \tag{10}$$

Results in Equation 9 and Equation 10 prove that the energy operation is a contraction. $\square$

**Theorem 2.** *Upon agent's convergence to an optimal policy $\pi^*$, total energy of $\pi^*$, expressed by $E^*$ will reach a thermal equilibrium consisting of minimum surprise among consecutive states $s$ and $s'$.*

*Proof.* We begin by initializing a set of $M$ policies $\{\pi_1, \pi_2..., \pi_M\}$ having energy ratios $\{E_1, E_2..., E_M\}$. Consider a policy $\pi_1$ with surprise value function $V_1$. $E_1$ can then be expressed as

$$E_1 = \log \left[\frac{\sum_{a=1}^{N} \exp\left(V_1^a(s', u', \sigma')\right)}{\sum_{a=1}^{N} \exp\left(V_1^a(s, u, \sigma)\right)}\right]$$

Invoking Assumption 2 for $s$ and $s'$, we can express $V_1^a(s', u', \sigma') = V_1^a(s, u, \sigma) + \zeta_1$ where $\zeta_1$ is a constant. Using this expression in $E_1$ we get,

$$E_1 = \log \left[\frac{\sum_{a=1}^{N} \exp\left(V_1^a(s, u, \sigma) + \zeta_1\right)}{\sum_{a=1}^{N} \exp\left(V_1^a(s, u, \sigma)\right)}\right]$$

$$E_1 = \log \left[\frac{\exp\left(\zeta_1\right) \sum_{a=1}^{N} \exp\left(V_1^a(s, u, \sigma)\right)}{\sum_{a=1}^{N} \exp\left(V_1^a(s, u, \sigma)\right)}\right]$$

$$E_1 = \zeta_1$$

Similarly, $E_2 = \zeta_2, E_3 = \zeta_3..., E_M = \zeta_M$. Thus, the energy residing in policy $\pi$ is proportional to the surprise between consecutive states $s$ and $s'$. Clearly, an optimal policy $\pi^*$ is the one with minimum surprise. Mathematically,

$$\pi^* \geq \pi_1, \pi_2..., \pi_M \implies \zeta^* \leq \zeta_1, \zeta_2..., \zeta_M$$
$$= \pi^* \geq \pi_1, \pi_2..., \pi_M \implies E^* \leq E_1, E_2..., E_M$$

Thus, proving that the optimal policy consists of minimum surprise at thermal equilibrium. $\qquad\square$

## B  Relation to Maximum Entropy Framework

### B.1  Similarities & Differences

We conceptually compare EMIX to the maximum entropy framework.

**Similarities:** Both methods utilize an auxilary objective as intrinsic motivation to tackle uncertainty. While the maximum entropy formulation assigns low energy to uncertain actions, our method assigns low energy to uncertain encoded representations od states (as presented in Fig. 2).

**Differences:** Our method differs from maximum entropy in its optimization process and learning scheme. The maximum entropy formulation aims to maximize entropy in the value function space so as to motivate exploration. Our proposed scheme, on the other hand, aims to minimize surprise in the low-dimensional representation space to obtain dynamics-aware robust policies.

### B.2  Connection to Soft Q-Learning

The Soft Q-Learning objective with $V^{\theta^-}_{\text{soft}}(s')$ and $Q_{\text{soft}}(u, s; \theta)$ as state and action value functions respectively is given by-

$$J_Q(\theta) = \mathbb{E}_{s,u \sim R} \left[ \frac{1}{2} \left( r + \gamma \mathbb{E}_{s' \sim R}[V^{\theta^-}_{\text{soft}}(s')] - Q_{\text{soft}}(u, s; \theta) \right)^2 \right]$$

$$= J_Q(\theta) = \mathbb{E}_{s,u \sim R} \left[ \frac{1}{2} \left( r + \gamma \mathbb{E}_{s' \sim R} \left[ \log \sum_{u \in A} \exp Q_{\text{soft}}(u', s'; \theta^-) \right] - Q_{\text{soft}}(u, s; \theta) \right)^2 \right]$$

The gradient of this objective can be expressed as-

$$\nabla_\theta J_Q(\theta) = \mathbb{E}_{s,u \sim R} \left[ \left( r + \gamma \mathbb{E}_{s' \sim R} \left[ \log \sum_{u \in A} \exp Q(u', s'; \theta^-) \right] - Q_{\text{soft}}(u, s; \theta) \right) \right] \nabla_\theta Q_{\text{soft}}(u, s; \theta)$$

(11)

And the gradient of the EMIX objective is obtained as-

$$L(\theta) = \mathbb{E}_{s,u,s' \sim R} \left[ \frac{1}{2} \left( r + \gamma \max_{u'} Q(u', s'; \theta^-) + \beta \log \left( \frac{\sum_{a=1}^N \exp\left(V^a_{\text{surp}}(s', u', \sigma')\right)}{\sum_{a=1}^N \exp\left(V^a_{\text{surp}}(s, u, \sigma)\right)} \right) - Q(u, s; \theta) \right)^2 \right]$$

$$\nabla_\theta L(\theta) = \mathbb{E}_{s,u,s' \sim R} \left[ \left( r + \gamma \max_{u'} Q(u', s'; \theta^-) \right. \right.$$
$$\left. \left. + \beta \log \left( \frac{\sum_{a=1}^N \exp\left(V^a_{\text{surp}}(s', u', \sigma')\right)}{\sum_{a=1}^N \exp\left(V^a_{\text{surp}}(s, u, \sigma)\right)} \right) - Q(u, s; \theta) \right) \right] \nabla_\theta Q(u, s; \theta) \quad (12)$$

Comparing Equation 11 to Equation 12 we notice that Soft Q-Learning and EMIX are related to each other as they utilize EBMs. Soft Q-Learning makes use of a discounted energy function which downweights the energy values over longer horizons. Actions consisting of lower energy configurations are given preference by making use of $Q_{\text{soft}}(u, s; \theta)$ as the negative energy. On the other hand, EMIX makes use of a constant energy function weighed by $\beta$ which minimizes surprise-based energy between consecutive states. Both the objectives can be thought of as energy minimizing models which search for an optimal energy configuration. Soft Q-Learning searches for an optimal configuration in the action space whereas EMIX favours optimal behavior on spurious states. In fact, EMIX can be realized as a special case of Soft Q-Learning if the mixer agent utilizes an energy-based policy and attains thermal equilibrium. This leads us to express Theorem 3.

**Theorem 3.** *Given an energy-based policy $\pi$ with its target function $V(s') = \log \sum_{u \in A} \exp Q(u', s'; \theta^-)$, the surprise minimization objective $L(\theta)$ reduces to the Soft Q-Learning objective $L(\theta_{\text{soft}})$ in the special case surprise absent between consecutive states, $\sum_{a=1}^{N} \exp\left(V_{\text{surp}}^a(s', u', \sigma')\right) = \sum_{a=1}^{N} \exp\left(V_{\text{surp}}^a(s, u, \sigma)\right)$.*

*Proof.* We know that the EMIX objective is given by-

$$L(\theta) = \mathbb{E}_{s,u,s' \sim R}\left[\frac{1}{2}\left(r + \gamma \max_{u'} Q(u'; s', \theta^-) + \beta \log\left(\frac{\sum_{a=1}^{N} \exp\left(V_{\text{surp}}^a(s', u', \sigma')\right)}{\sum_{a=1}^{N} \exp\left(V_{\text{surp}}^a(s, u, \sigma)\right)}\right) - Q(u, s; \theta)\right)^2\right]$$

(13)

Replacing the greedy policy term $\max_{u'} Q(u', s'; \theta^-)$ with the energy-based value function $V(s') = \log \sum_{u' \in A} \exp Q(u', s'; \theta^-)$, we get,

$$L(\theta) = \mathbb{E}_{s,u,s' \sim R}\left[\frac{1}{2}\left(r + \gamma \mathbb{E}_{s' \sim R}[V(s')] + \beta \log\left(\frac{\sum_{a=1}^{N} \exp\left(V_{\text{surp}}^a(s', u', \sigma')\right)}{\sum_{a=1}^{N} \exp\left(V_{\text{surp}}^a(s, u, \sigma)\right)}\right) - Q(u, s; \theta)\right)^2\right]$$

(14)

$$= L(\theta) = \mathbb{E}_{s,u,s' \sim R}\left[\frac{1}{2}\left(r + \gamma \mathbb{E}_{s' \sim R}\left[\log \sum_{u' \in A} \exp Q(u', s'; \theta^-)\right]\right.\right.$$
$$\left.\left. + \beta \log\left(\frac{\sum_{a=1}^{N} \exp\left(V_{\text{surp}}^a(s', u', \sigma')\right)}{\sum_{a=1}^{N} \exp\left(V_{\text{surp}}^a(s, u, \sigma)\right)}\right) - Q(u, s; \theta)\right)^2\right]$$

At thermal equilibrium, $\sum_{a=1}^{N} \exp\left(V_{\text{surp}}^a(s, u, \sigma)\right) = \sum_{a=1}^{N} \exp\left(V_{\text{surp}}^a(s', u', \sigma')\right)$,

$$= L(\theta) = \mathbb{E}_{s,u,s' \sim R}\left[\frac{1}{2}\left(r + \gamma \mathbb{E}_{s' \sim R}\left[\log \sum_{u' \in A} \exp Q(u', s'; \theta^-)\right]\right.\right.$$
$$\left.\left. + \beta \log\left(\frac{\sum_{a=1}^{N} \exp\left(V_{\text{surp}}^a(s', u', \sigma')\right)}{\sum_{a=1}^{N} \exp\left(V_{\text{surp}}^a(s', u', \sigma')\right)}\right) - Q(u, s; \theta)\right)^2\right]$$

$$= L(\theta) = \mathbb{E}_{s,u,s' \sim R}\left[\frac{1}{2}\left(r + \gamma \mathbb{E}_{s' \sim R}\left[\log \sum_{u' \in A} \exp Q(u', s'; \theta^-)\right] + \beta \log(1) - Q(u, s; \theta)\right)^2\right]$$

(15)

$$= L(\theta) = \mathbb{E}_{s,u,s' \sim R}\left[\frac{1}{2}\left(r + \gamma \mathbb{E}_{s' \sim R}\left[\log \sum_{u' \in A} \exp Q(u', s'; \theta^-)\right] - Q(u, s; \theta)\right)^2\right] \qquad (16)$$

Equation 16 represents the Soft Q-Learning objective, hence proving the result. $\qquad \square$

## C   Convergence Analysis

We now analyze convergence of the surprise minimization scheme during policy optimization. Our notation denotes $\mathcal{B}V_{k-1} = r + \gamma V_{k-1}$ as the Bellman operator which obeys monotonicity and contraction.

Monotonicity: $V_1 \leq V_2 \implies \mathcal{B}V_1 \leq \mathcal{B}V_2$ ;   Contraction: $\|\mathcal{B}V_1 - \mathcal{B}V_2\|_2 \leq \gamma \|V_1 - V_2\|_2$

We now denote $\hat{V}_k = r_k + \gamma \hat{V}_{k-1} + \beta \log \sum_{a=1}^{N} \exp(V_{\text{surp},(k)}^a(s, u, \sigma))$ as the total value at step $k$. As per the definition of $\mathcal{B}$, this gives us $\hat{V}_k = \mathcal{B}\hat{V}_{k-1} + \beta \log \sum_{a=1}^{N} \exp(V_{\text{surp},(k)}^a(s, u, \sigma))$.

Consider $\left\|\hat{V}_k - V^*\right\|_2$ with $V^*$ being the optimal value at convergence,

$$\left\|\hat{V}_k - V^*\right\|_2 \leq \left\|\mathcal{B}\hat{V}_{k-1} + \beta \log \sum_{a=1}^{N} \exp(V_{\text{surp},(k)}^a) - V^*\right\|_2 \tag{17}$$

Where $\beta \log \sum_{a=1}^{N} \exp(V_{\text{surp},(k)}^a) > 0$ as per constant positive surprise $\zeta > 0$ in 2. We impose this constraint by adding ReLU nonlinearities in surprise encoder to obtain positive $V_{\text{surp},(k)}^a$ values.

$$\leq \left\|\mathcal{B}^2\hat{V}_{k-2} + \beta \log \sum_{a=1}^{N} \exp(V_{\text{surp},(k-1)}^a) + \beta \log \sum_{a=1}^{N} \exp(V_{\text{surp},(k)}^a) - V^*\right\|_2 \tag{18}$$

$$\leq \left\|\mathcal{B}^2\hat{V}_{k-2} + \beta \left( \log \sum_{a=1}^{N} \exp(V_{\text{surp},(k-1)}^a) + \log \sum_{a=1}^{N} \exp(V_{\text{surp},(k)}^a) \right) - V^*\right\|_2 \tag{19}$$

$$\leq \left\|\mathcal{B}^2\hat{V}_{k-2} + \beta \left( \log \left[\sum_{a=1}^{N} \exp(V_{\text{surp},(k-1)}^a)\right]\left[\sum_{a=1}^{N} \exp(V_{\text{surp},(k)}^a)\right] \right) - V^*\right\|_2 \tag{20}$$

Thus, for $k$ iterations, we have,

$$\leq \left\|\mathcal{B}^k V_0 + \beta \left( \log \prod_{i=1}^{k} \left[\sum_{a=1}^{N} \exp(V_{\text{surp},(i)}^a)\right] \right) - V^*\right\|_2 \tag{21}$$

$$= \left\|\mathcal{B}^k V_0 + \beta \left( \log \sum_{a=1}^{N} \left[\prod_{i=1}^{k} \exp(V_{\text{surp},(i)}^a)\right] \right) - V^*\right\|_2 \tag{22}$$

$$= \left\|\mathcal{B}^k V_0 + \beta \left( \log \sum_{a=1}^{N} \left[\exp(\sum_{i=1}^{k} V_{\text{surp},(i)}^a)\right] \right) - V^*\right\|_2 \tag{23}$$

We now absorb the sum of surprise values from time index $i = 1,..,k$ in a single variable $V_{\text{tot}}^a$. Thus, using $V_{\text{tot}}^a = \sum_{i=1}^{k} V_{\text{surp},(i)}^a$ and utilizing the Triangle Inequality, we get,

$$= \left\|\mathcal{B}^k V_0 - V^*\right\|_2 + \left\|\beta \left( \log \sum_{a=1}^{N} [\exp(V_{\text{tot}}^a)] \right)\right\|_2 \tag{24}$$

We now bound the two terms separately. Considering the first term and following the results of value iteration convergence [5],

$$\left\|\mathcal{B}^k V - V^*\right\|_2 \leq \gamma^k \left\|V - V^*\right\|_2 \tag{25}$$

$$\left\|\mathcal{B}^k V_0 - V^*\right\|_2 \leq \gamma^k \left\|V + V_\mu - V_\mu - V^*\right\|_2 \tag{26}$$

wherein $V_\mu$ denotes an approximation to $V$. Utilizing the triangle inequality yields,

$$\left\|\mathcal{B}^k V_0 - V^*\right\|_2 \leq \gamma^k \left\|V - V_\mu\right\|_2 + \gamma^k \left\|V_\mu - V^*\right\|_2 \tag{27}$$

The two terms are bounded using the convergence result of [4].

$$\left\|\mathcal{B}^k V_0 - V^*\right\|_2 \leq \gamma^k \sqrt{r_{\max}} + \gamma^k \sqrt{\frac{r_{\max}|\mathcal{S}|}{1 - \gamma}} \tag{28}$$

Now, considering the second term in Equation 24 and denoting $V_{\text{tot}}^* = \sum_{i=1}^{k} V_{\text{surp},(i)}^*$ as the sum of optimal surprise values,

$$\beta \left\|\log \sum_{a=1}^{N} \exp(V_{\text{tot}}^a)\right\|_2 = \beta \left\|\log \sum_{a=1}^{N} \exp(V_{\text{tot}}^a) - \log \sum_{a=1}^{N} \exp(V_{\text{tot}}^*) + \log \sum_{a=1}^{N} \exp(V_{\text{tot}}^*)\right\|_2 \tag{29}$$

using the triangle inequality,

$$\leq \beta \left\| \log \sum_{a=1}^{N} \exp(V_{\text{tot}}^a) - \log \sum_{a=1}^{N} \exp(V_{\text{tot}}^*) \right\|_2 + \beta \left\| \log \sum_{a=1}^{N} \exp(V_{\text{tot}}^*) \right\|_2 \tag{30}$$

Since $\mathcal{T} = \log \sum_{a=1}^{N} \exp(V_{\text{tot}}^a)$ is a contraction following Theorem 1, for the first term we have,

$$\leq \beta\gamma \left\| V_{\text{tot}}^a - V_{\text{tot}}^* \right\|_2 + \beta \left\| \log \sum_{a=1}^{N} \exp(V_{\text{tot}}^*) \right\|_2 \tag{31}$$

The second term in the above relation is bounded due to the completeness assumption, $\left\| \log \sum_{a=1}^{N} \exp(V_{\text{tot}}^*) \right\|_2$. The first term, on the other hand, is simplified by applying Jensen's Inequality on the definitions of $V_{\text{tot}}^a$ and $V_{\text{tot}}^*$, $\left\| \sum_{i=1}^{k} V_{\text{surp},(i)}^a - \sum_{i=1}^{k} V_{\text{surp},(i)}^* \right\|_2 = \left\| \sum_{i=1}^{k} \left( V_{\text{surp},(i)}^a - V_{\text{surp},(i)}^* \right) \right\|_2 \leq \sum_{i=1}^{k} \left\| V_{\text{surp},(i)}^a - V_{\text{surp},(i)}^* \right\|_2$. Denoting $\left\| V_{\text{surp},(i)}^a - V_{\text{surp},(i)}^* \right\|_2 = \text{RMSE}(V_{\text{surp},(i)}^a)$, we obtain the following result,

$$\leq \beta\gamma \sum_{i=1}^{k} \text{RMSE}(V_{\text{surp},(i)}^a) + \beta\zeta \ , \ \zeta > 0 \tag{32}$$

Finally, combining Equation 28 and Equation 32 in Equation 24, we obtain the desired convergence bound.

$$\left\| V_k - V^* \right\|_2 \leq \gamma^k \left( \sqrt{r_{\max}} + \sqrt{\frac{r_{\max}|\mathcal{S}|}{1-\gamma}} \right) + \beta \left( \gamma \sum_{i=1}^{k} \text{RMSE}(V_{\text{surp},(i)}^a) + \zeta \right) \tag{33}$$

While the first term in Equation 33 denotes the convergence of policy optimization, the second term indicates the bounded convergence of surprise to ecological niches with finite (yet nonzero) surprising elements. The policy optimization process converges at a geometric rate $\mathcal{O}(\gamma^k)$ towards its stable fixed points. The surprise minimization process, on the other hand, demonstrates an annealing behavior which depends on the temperature parameter $\beta$. Furthermore, convergence to stable fixed point $V_{\text{tot}}^a$ is bounded in respect to each agents individual surprise values $V_{\text{tot}}^a$. This insight indicates that different agents converge towards different locally optimal values of surprise. Finally, the presence of constant $\zeta$ corroborates prior claims [50, 12] that agents continue to experience surprise irrespective of their convergence to minimum energy niches. To further develop intuition for this claim, consider the special case wherein $\sum_{i=1}^{k} \text{RMSE}(V_{\text{surp},(i)}^a) \to 0$, i.e.- surprise estimation error for all iterations goes to zero. Irrespective of global convergence among all agents, a finite yet small $\zeta$ continues to contribute to the upper bound of $\left\| V_k - V^* \right\|_2$.

**Role of $\beta$:** We further discuss the role of $\beta$ which is of balancing the terms at successive iterations. While the first term geometrically decays with $\mathcal{O}(\gamma^k)$ rate, the second term approaches a finite constant $\beta\zeta$ as $V_{tot}^a \to V_{tot}^*$. Irrespective of our choice of $\beta$, the LHS $\|V_k - V^*\|_2$ is upper bounded by a constant which validates the claims of minimum yet finite surprise values. We do note that a small $\beta$ is still desirable to remove any approximation errors in order to push $V_k \to V^*$. However, this comes at the cost of increased surprise if $\beta$ is not selected appropriately.

## D Implementation Details

### D.1 Model Specifications

**Architecture:** This section highlights model architecture for the surprise value function. At the lower level, the architecture consists of 3 independent networks called *state_net*, *q_net* and *surp_net*.

Each of these networks consist of a single layer of 256 units with ReLU non-linearity as activations. Similar to the mixer-network, we use the ReLU non-linearity in order to provide monotonicity constraints across agents. Using a modular architecture in combination with independent networks leads to a richer extraction of joint latent transition space. Outputs from each of the networks are concatenated and are provided as input to the *main_net* consisting of 256 units with ReLU activations. The *main_net* yields a single output as the surprise value $V_{\mathrm{surp}}^{a}(s, u, \sigma)$ which is reduced along the agent dimension by the energy operator. Alternatively, deeper versions of networks can be used in order to make the extracted embeddings increasingly expressive. However, increasing the number of layers does little in comparison to additional computational expense.

**Computation of $\sigma$:** The deviation $\sigma$ corresponds to the standard deviation across each dimension of the state $s$. Considering the state as a tensor of size $B \times A \times M$ with $B$ as the batch size, $A$ as the number of agents and $M$ as the observation dimension, we compute $\sigma$ by calculating the standard deviation across the $M$ dimension. This yields $\sigma$ as a $B \times A \times 1$ dimensional array.

**Computation of surprise estimates:** $V_{\mathrm{surp}}$ denotes the surprise value function which quantifies the amount of surprise experienced by agents. Analogous to a $Q$ value function which provides estimates of returns, $V_{\mathrm{surp}}$ provides estimate of surprise. Our framework learns $V_{surp}$ much like any other value function (using a neural network), but by additionally undergoing a $\log \sum \exp$ transformation to obey the fixed point property. This is achieved by realizing log-sum-exp as an energy operator $\mathcal{T} = \log \sum \exp$ which can be computed using standard computation libraries. Since our code is implemented in PyTorch, we implement this as `T_V = torch.logsumexp(V_surp, dim=1)`.

**Global State Encoder:** The global state encoder serves as a mapping from the state space to a low dimensional representation space $\mathcal{S} \to \mathcal{Z}$. The encoder takes in a sequence of states $\{s_1, s_2, ..., s_T\}$ as input and outputs a latent representation $z_{\mathrm{state}}$. We use a standard pyramid MLP network consisting of 2 hidden layers of 256 units each with ReLU non-linearity. Embeddings obtained from the encoder are concatenated with other latent embeddings before being passed to the final surprise encoder.

**Standard Deviation Encoder:** The standard deviation encoder serves as a mapping from standard deviations across state dimensions to a low dimensional representation space. Each standard deviation $\sigma$ is computed across dimensions of the state $s_t$. These deviations are then packed in a sequence $\{\sigma_1, \sigma_2, ..., \sigma_T\}$ and passed as inputs to the standard deviation encoder. Intuitively, the encoder learns changes across states in a batch of observations. This is similar to a dynamics model predicting future states, except that we map these states to a low dimensional embedding. We use a standard pyramid MLP network consisting of 2 hidden layers of 256 units each with ReLU non-linearity. Embeddings obtained from the encoder are concatenated with other latent representations and used by the final surprise encoder to estimate the surprise distribution.

## D.2 Hyperparameters

Table 3 presents hyperparameter values for EMIX. A total of 2 target $Q$-functions were used as the model is found to be robust to any greater values.

| Hyperparameters | Values |
|---|---|
| batch size | $b = 32$ |
| learning rate | $\alpha = 0.0005$ |
| discount factor | $\gamma = 0.99$ |
| target update interval | 200 episodes |
| gradient clipping | 10 |
| exploration schedule | 1.0 to 0.01 over 50000 steps |
| mixer embedding size | 32 |
| agent hidden size | 64 |
| temperature | $\beta = 0.01$ |
| target $Q$-functions | 2 |

Table 3: Hyperparameter values for EMIX agents

### D.3 Selection & Tuning of $\beta$

One can manually tune $\beta$ using a fine-grained hyperparameter search. We tune $\beta$ between 0.001 and 1 in intervals of 0.01 with best performance observed at $\beta = 0.01$. However, we find two additional methods helpful for obtaining more accurate values. These are described as follows-

**Armijo's Line Search:** One can borrow from optimization theory and utilize Armijo's line search [41] by setting a termination condition. The method starts with a constant value of $\beta$ which is iteratively incremented/decremented until a termination criterion (example- $\|\nabla L(\theta)\| < \epsilon$ with $\epsilon$ a constant) is reached. While line search is proven to converge towards globally optimal values, its $\mathcal{O}(n^2)$ convergence may be computationally expensive that too in the MARL setting. Thus, we turn to the more efficient automatic tuning.

---

**Algorithm 2** Armijo's Line Search

1: Initialize $\beta$, $\delta \in (0, 1]$, EMIX & $\mathcal{TV}_{\text{surp}}^a$;
2: **while** EMIX$(Q + \beta * \mathcal{TV}_{\text{surp}}^a) >$ EMIX$(Q)$
  $+ \alpha * \beta * \nabla$EMIX$(\mathcal{Q})^{\text{T}}\mathcal{TV}_{\text{surp}}^a$ **do**
3:   $\beta = \delta * \beta$
4: **end while**
5: return $\beta$

---

**Algorithm 3** Automatic Tuning

1: Initialize $\beta$, $\delta \in (0, 1]$, EMIX & $\mathcal{TV}_{\text{surp}}^a$;
2: EMIX$(Q + \beta * \mathcal{TV}_{\text{surp}}^a)$
3: beta_loss $= \beta * 0.5 * (\mathcal{TV}_{\text{surp}}^a - 0)^2$
4: beta_loss.backward()
5: return $\beta$

---

**Automatic Tuning:** We choose to automatically tune $\beta$ following single-agent RL literature [20, 28]. This is achieved by treating $\beta$ as a parameter and adaptively optimizing over it using Adam. We treat a surprise value of 0 as our target value. The method works well in practice and provides $\beta$ values closer to 0.01 (our manual selection).

## E   Additional Results

### E.1   Statistical Significance

| Scenarios | EMIX | SMiRL-QMIX | QMIX | VDN | COMA | IQL |
|-----------|------|------------|------|-----|------|-----|
| 2s_vs_1sc | 14 | 7 | - | 21 | **25** | 4 |
| 2s3z | **15** | 9 | - | 6 | 0 | 0 |
| 3m | **17** | 0 | - | 0 | 2 | 12 |
| 3s_vs_3z | **11** | 3 | - | 0 | 0 | 1 |
| 3s_vs_4z | **21** | 0 | - | 2 | 0 | 0 |
| 3s_vs_5z | 5 | 0 | - | **25** | 0 | 0 |
| 3s5z | 7 | **13** | - | 0 | 0 | 0 |
| 8m | **15** | 1 | - | 1 | 3 | 0 |
| 8m_vs_9m | 7 | **11** | - | 0 | 0 | 0 |
| 10m_vs_11m | 14 | **25** | - | 6 | 0 | 0 |
| so_many_baneling | **24** | 14 | - | 9 | 4 | 0 |
| 5m_vs_6m | **21** | 15 | - | 18 | 0 | 0 |

Table 4: Comparison of the $\mathcal{U}$ statistic on StarCraft II benchmark. $\mathcal{U}$ here denotes the statistical significance of an algorithm against QMIX (higher is better).

We follow the recommendation of [33] and evaluate the statistical significance of our results by carrying out the Mann-Whitney U test [38]. All 5 seeds of an algorithm (on each task) are compared to that of QMIX to yield the $\mathcal{U}$ statistic. $\mathcal{U}$ here denotes the statistical significance of performance with higher values being desirable.

Table 4 presents the comparison of $\mathcal{U}$ statistic on the StartCraft II benchmark. EMIX demonstrates consistently high values of $\mathcal{U}$ across a diverse set of tasks when compared to SMiRL and prior MARL agents. This highlights the consistent surprise-minimizing performance of EMIX across random seeds.

### E.2 StarCraft II Benchmark

| Scenarios | EMIX | SMiRL-QMIX | QMIX | VDN | COMA | IQL |
|---|---|---|---|---|---|---|
| 2s_vs_1sc | $90.33 \pm 0.72$ | $88.41 \pm 1.31$ | $89.19 \pm 3.23$ | $91.42 \pm 1.23$ | $\mathbf{96.90 \pm 0.54}$ | $86.07 \pm 0.98$ |
| 2s3z | $\mathbf{95.40 \pm 0.45}$ | $94.93 \pm 0.32$ | $\mathbf{95.30 \pm 1.28}$ | $92.03 \pm 2.08$ | $43.33 \pm 2.70$ | $55.74 \pm 6.84$ |
| 3s_vs_3z | $\mathbf{99.58 \pm 0.07}$ | $97.63 \pm 1.08$ | $\mathbf{99.43 \pm 0.20}$ | $97.90 \pm 0.58$ | $0.21 \pm 0.54$ | $92.32 \pm 2.83$ |
| 3s_vs_5z | $52.91 \pm 11.80$ | $0.00 \pm 0.00$ | $43.44 \pm 7.09$ | $\mathbf{68.51 \pm 5.60}$ | $0.00 \pm 0.00$ | $18.14 \pm 2.34$ |
| 3s5z | $\mathbf{88.88 \pm 1.07}$ | $88.53 \pm 1.03$ | $\mathbf{88.49 \pm 2.32}$ | $63.58 \pm 3.99$ | $0.25 \pm 0.11$ | $7.05 \pm 3.52$ |
| 8m | $\mathbf{94.47 \pm 1.38}$ | $89.96 \pm 1.42$ | $\mathbf{94.30 \pm 2.90}$ | $90.26 \pm 1.12$ | $92.82 \pm 0.53$ | $83.53 \pm 1.62$ |

Table 5: Comparison of success rate percentages between EMIX and prior MARL methods on StarCraft II micromanagement scenarios. EMIX is comparable to or improves over QMIX agent. In comparison to SMiRL-QMIX, EMIX demonstrates improved minimization of surprise. Results are averaged over 5 random seeds.

### E.3 Predator-Prey Benchmark

We consider a simple toy task from the Predator-Prey benchmark to demonstrate the importance of surprise minimization. We select *predator_prey_easy* due to its simplicity and convenient dynamics. The task consists of 3 agents and 3 opponents. We increase the number of opponents while keeping the task fixed. This way the dynamics of the MDP remain unchanged and the only changing factor is opponent behaviors.

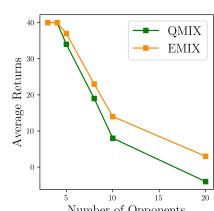

Fig. 8 presents the variation of average returns for EMIX and QMIX over 5 random seeds. While QMIX agents undergo a steady decrease in performance, EMIX agents are found robust to this fast degradation. Even after the addition of 20 opponents (against only 3 agents), EMIX is able to retain positive returns. The algorithm acquires a surprise robust-policy early on during training to tackle fast-paced changes introduced by the large number of agents.

Figure 8: Variation in performance with increasing number of agents.

### E.4 Note on Minimum Entropy Conjugate Objective

The minimum conjugate entropy objective denotes the dual problem to surprise minimization. If we compute the Legendre Transform of our energy-based operator $\mathcal{T}V^a_{\mathrm{surp}}(s, u, \sigma) = \log \sum_{a=1}^{N} \exp(V^a_{\mathrm{surp}}(s, u, \sigma))$ we obtain the entropy function $\mathcal{H}(x)$ where x is the gradient of the operator, $x = \mathcal{T}V^a_{\mathrm{surp}}(s, u, \sigma)$. This insight indicates that minimizing the energy operator $\mathcal{T}V^a_{\mathrm{surp}}(s, u, \sigma)$ is same as minimizing entropy in the space of gradients. Intuitively, our objective aims to minimize uncertainty in the learning signal.

## F   Additional Related Work on Multi-Agent Value Factorization

We discuss recent MARL methods within the Centralised Training and Decentralised Control paradigm [26] which improve value factorization. The original work of QTRAN [51] improves representational capacity of factorization schemes by generalizing methods such as QMIX [44] and VDN [53]. More recent advances combine techniques from dueling networks and temporal abstraction to learn MARL agent factorizations with sufficient representations [59]. A notable work is that of [60] which employs pretrained action representations to learn agent-specific roles. Decomposing policy optimization into role selection and role execution stages allows larger number of MARL agents to collaborate well even in unseen scenarios. Alternate works consider information theoretic objectives to introduce diversity in optimization and representation of shared multi-agent parameters [10]. Lastly, [61] extend these ideas by decomposing value learning in multi-agent actor-critic methods. Within the off-policy setting, these agents highlight sufficient representational capacity in both discrete and continuous action spaces.