# OpenReview forum: "Surprise Minimizing Multi-Agent Learning with Energy-based Models"
_NeurIPS.cc/2022/Conference — NeurIPS 2022 Accept_

### Official Review · Reviewer_RyKq · 2022-07-04

**Rating:** 6
**Confidence:** 5
**Soundness:** 3 good
**Presentation:** 3 good
**Contribution:** 2 fair

**Summary:**

This paper proposed incorporating surprise minimization in multi-agent reinforcement learning (MARL) through temporal energy models for capturing the uncertainty across agents. More specifically, this paper formulated an energy function extending the energy-based model theory and proposed an operator to temproally update this energy function. Moreover, this paper analysed the relationship to the soft Q-learning and gave a convergence analysis for the proposed method. With practical function approximation, the proposed algorithm according to the theory called EMIX was shown achieving superior performance with tiny margins compared with the baselines.

**Questions:**

1. Assumption 2 needs more explanation on why the termination state appears in the discounted problem settings.
2. I wish the authors give the details on how Eq. (17) is obtained.

**Limitations:**

The authors gave the limitations and the future work to address it.

**Strengths And Weaknesses:**

**Strengths:**
1. The presentation of this paper is good, starting with a reasonable question as the motivation and addresing it by a complete methodology.
2. The theoretical proofs are generally correct.
3. The related works discussed in this paper is sufficient.
4. It compared the relation to maximum entropy framework.

**Weaknesses:**
1. Assumption 2 is inconsistent with the discounted problem considered in this paper. To my best knowledge, there does not exist a termination state in the context of discounted problems. Nevertheless, it exists in the stochastic shortest path (SSP) problem.
2. The experimental results of baselines could be underestimated compared with the previous works.

---

> ### Author Response · Authors · 2022-08-01
> **Response to Reviewer RyKq**
>
> We thank the respected reviewer for providing detailed feedback on the paper which is of utmost value to our work. We address your concerns below and in the general response comment above.
>
> **Assumption 2 needs more explanation on why the termination state appears in the discounted problem settings.**
> Our setting considers the finite-horizon discounted problem, or Dec-POMDP [1], i.e- episodes end at the termination state $s\_{T}$ at timestep T. We have added the above statement in Section 3. In the case of infinite-horizon discounted problems [2], a termination state would remain absent and Assumption 2 would not hold. However, we follow prior MARL works and consider the finite-horizon framework. Assumption 2 bounds surprise between successive states. Since our setting is episodic with a finite-horizon, the episode will terminate after $T$ timesteps. Thus, reward and surprise incurred by agents are always bounded over an episode.
>
> **The experimental results of baselines could be underestimated compared with the previous works.**
> We emphasize that EMIX agents maintain competitive results to prior MARL methods while minimizing surprise. Additionally, our energy-based scheme is found more favourable over the SMiRL method which is currently the only baseline for surprise minimization in literature. Our consistent returns over 5 random seeds also show that EMIX is capable of expanding this performance towards other benchmarks and downstream applications, an area where previous MARL methods have struggled. Finally, we also highlight our qualitative results in Appendix E.1 which demonstrate that EMIX agents retrieve robust policies at test-time, a phenomenon seldomly observed in QMIX agents.
>
> **I wish the authors give the details on how Eq. (17) is obtained.**
> Thank you for the suggestion. We have added additional details on Bellman operator, its properties and its usage in obtaining Equation 17.
>
> Our notation denotes $\mathcal{B} V\_{k-1} = r + \gamma V\_{k-1}$ as the Bellman operator which obeys monotonicity and contraction.
>
> Monotonicity: $V\_{1} \leq V\_{2} \implies \mathcal{B} V\_{1} \leq \mathcal{B} V\_{2}$
>
> Contraction: $\| \mathcal{B}V\_{1} - \mathcal{B}V\_{2} \|\_{2} \leq \gamma \| V\_{1} - V\_{2} \|\_{2}$
>
> We now denote $\hat{V}\_{k} = r\_{k} + \gamma \hat{V}\_{k-1} + \beta \log \sum\_{a=1}^{N} \exp (V^{a}\_{surp,(k)}(s,u,\sigma))$ as the total value at step $k$. As per the definition of $\mathcal{B}$, this gives us $\hat{V}\_{k} = \mathcal{B} \hat{V}\_{k-1} + \beta \log \sum\_{a=1}^{N} \exp (V^{a}\_{surp,(k)}(s,u,\sigma))$.
>
> We use the above step to obtain Equation 17 and continue our calculation.
>
> Kindly let us know if our response above addresed your concerns.
>
> [1]. F. A. Oliehoek and C. Amato. A concise introduction to decentralized POMDPs. Springer, 2016.
>
> [2]. R. Sutton, A. Barto, Reinforcement Learning: An Introduction, MIT Press, 2018.

---

> > ### Comment · Reviewer_RyKq · 2022-08-03
> > **Response to Authors**
> >
> > Thanks for the authors' response which have  addressed most of my concerns. However, the last one is still unsolved.
> >
> > My main concern is on the sign of $\beta \log \sum_{a=1}^{N} \exp ( ... )$. If its sign is negative (that could happen), it seems that Eq. 17 does not necessarily hold.

---

> > > ### Author Response · Authors · 2022-08-04
> > > **Response to Follow-up**
> > >
> > > Thank you for the follow up question! Our Assmuption 2 handles this case theoretically by explicitly restricting our analysis to settings of finite and positive true surprise ($\zeta > 0$). Additionally, we empirically impose this constraint in our architecture by using ReLU nonlinearity at the last layer of surprise encoder when outputting $V^{a}_{\text{surp}}(s,u,\sigma)$. This makes $\beta \log \sum \exp$ always yield a positive value.

---

> > > > ### Comment · Reviewer_RyKq · 2022-08-04
> > > > **Follow-up Response to Authors**
> > > >
> > > > Thanks for the authors' further response that has addressed my concerns. Nevertheless, I suggest the authors add these to the proof to avoid the misunderstanding from the audience.

---

> > > > > ### Author Response · Authors · 2022-08-04
> > > > > **Follow-up and changes**
> > > > >
> > > > > Thank you for the suggestion. We have added the following discussion in lines 581-582 to explicitly highlight positivity of the surprise minimization term-
> > > > >
> > > > > Where $\beta \log \sum\_{a=1}^{N} \exp(V^{a}\_{surp,(k)}) > 0$ as per constant positive surprise $\zeta > 0$ in Assumption 2. We impose this constraint by adding ReLU nonlinearities in surprise encoder to obtain positive $V^{a}_{surp,(k)}$ values.

---

### Official Review · Reviewer_DDcB · 2022-07-09

**Rating:** 7
**Confidence:** 5
**Soundness:** 3 good
**Presentation:** 2 fair
**Contribution:** 3 good

**Summary:**

Paper introduces a new algorithm within the paradigm of Centralised Training and Decentralised Execution for multi-agent games. The authors introduce a new method EMIX - which improves upon QMIX by adding a global surprise minimisation objective to the centralised-value function training. The energy-based model (EBM) approximation is shown to provide stability during training through ablations.


The paper provides strong theoretical justification that 1) the surprise operator converges to a fixed-point, and 2) that as the policy converges to an optimal policy then surprise is also minimised. The paper also demonstrates improved performance than baselines on most games within Starcraft II and through ablations demonstrates the necessity for both EBM approximations and surprise reward shaping.

**Questions:**

1. Could the authors provide any more indepth analysis of how this outperforms COMA and QMIX, the analysis given in E1 is exceptionally useful and would be appreciated if more could be given of this flavour. I’d even recommend bringing more of this into the main paper.

2. The constant beta is not tuned at all during training - similar to SAC could you not train this?

3. The global state encoder and standard deviation encoders are a slight mystery to me and I’d like more understanding of how they are set up and trained.

4. It is unclear how the V_surp is learnt or obtained. What is the surprise method trained on, how is this estimated during training?


**Limitations:**

The authors have discussed the limitations of this approach as it is scaled.

**Strengths And Weaknesses:**

Strengths:

Strong theoretical justification

Clear Presentation

High quality paper

Seems like an original extension in the Centralised Training Decentralised Deployment algorithm family.

Weaknesses:

The global surprise encoder could be given slightly more description. The diagram is good but an additional example of the flow for interference or training optimisation would be appreciated.

Performance is only marginally better  (a more challenging benchmark might be necessary). This is the most unconvincing component of the paper but i do appreciate the limited benchmarks available currently.

---

> ### Author Response · Authors · 2022-08-01
> **Response to Reviewer DDcB (2/2)**
>
>
> **It is unclear how the V\_surp is learnt or obtained. What is the surprise method trained on, how is this estimated during training?**
> Our framework utilizes $E$ as a loss function to contrastively learn $V^{a}\_{surp}(s,u,\sigma)$ during training (line 13 in Algorithm 1). $E$ acts as intrinsic motivation as well as a learning signal for updating $V^{a}\_{surp}(s,u,\sigma)$ estimates. We use the architecture of Figure 2 to compute a possible distribution of surprise. We then sample from this distribution using the reparameterization trick. Our framework learns $V^{a}\_{surp}(s,u,\sigma)$ much like any other value function (using a neural network), but by additionally undergoing the energy operator transformation $\mathcal{T} V^{a}\_{surp}(s,u,\sigma)$.
>
> Kindly let us know if our response above addresed your concerns.

---

> > ### Comment · Reviewer_DDcB · 2022-08-06
> > **Thanks for the clarifications.**
> >
> > Thanks for addressing my comments! No further questions

---

> ### Author Response · Authors · 2022-08-01
> **Response to Reviewer DDcB (1/2)**
>
> We thank the respected reviewer for providing detailed feedback on the paper which is of utmost value to our work. We address your concerns below and in the general response comment above.
>
> **Performance is only marginally better.**
> We emphasize the importance of EMIX is highlighted by its surprise mininizing objective. Compared to prior surprise minimization schemes such as SMiRL-QMIX, EMIX consistently improves performance on all tasks. When compared to prior MARL methods, EMIX retains optimal returns. Qualitatively, EMIX agents produce robust policies which balance between task returns and safety of agents. The key contribution of EMIX lies in its suitability to maintain optimal performance **while** minimizing surprise, producing robust policies and stabilizing training at lower $\beta$ temperature values. We believe this to be an important stepping stone towards surprise-robust MARL agents.
>
> **Could the authors provide any more in-depth analysis of how this outperforms COMA and QMIX, the analysis given in E1 is exceptionally useful and would be appreciated if more could be given of this flavour. I’d even recommend bringing more of this into the main paper.**
> Thank you for the suggestion. EMIX demonstrates qualitatively robust policies when compared to QMIX agents. Additionally, EMIX is found robust to increasing number of agents and strict reward penalties on the Predator-Prey benchmark (Appendix E.4). Following your suggestion, we aim to move these qualitative results in the main text in the final version. Qualitative evaluation of MARL agents is still an active area of research. We continue to run more experiments and observe relevant emergent behaviors on other taks.
>
> **The constant beta is not tuned at all during training - similar to SAC could you not train this?**
> We tuned $\beta$ during training but found it to provide little to no improvements on task returns. Appendix D.3 provides a description of this procedure. In practice, we may utilize two methods to select and tune $\beta$. These are described as follows-
>
> **(1) Armijo's Line Search-** One can borrow from optimization theory and utilize Armijo's line search with termination conditions. The method starts with a constant value of  which is iteratively incremented/decremented until a termination criterion (example-  with  a constant) is reached. While line search is proven to converge towards globally optimal values, it turns out to be computationally expensive. Thus, we turn to the more efficient automatic tuning.
>
> **(2) Automatic Tuning-** We choose to automatically tune $\beta$ following single-agent RL literature. This is achieved by treating $\beta$ as a parameter and adaptively optimizing over it using Adam. We treat a surprise value of $0$ as our target value. The method works well in practice and provides $\beta$ values closer to $0.01$ (our manual selection).
>
> **The global state encoder and standard deviation encoders are a slight mystery to me and I’d like more understanding of how they are set up and trained.**
> Thank you for the suggestion. We have added additional discussion on global state encoder and standard deviation encoders in Appendix D.1 for the timebeing. We aim to move this to the main text in the final version. The discussion is summarised below for conciseness-
>
> **Global State Encoder-** The global state encoder serves as a mapping from the state space to a low dimensional representation space $\mathcal{S} \rightarrow \mathcal{Z}$. The encoder takes in a sequence of states $\{ s\_{1}, s\_{2}, ..., s\_{T} \}$ as input and outputs a latent representation $z\_{\text{state}}$. We use a standard pyramid MLP network consisting of 2 hidden layers of 256 units each with ReLU non-linearity. Embeddings obtained from the encoder are concatenated with other latent embeddings before being passed to the final surprise encoder.
>
> **Standard Deviation Encoder-** The standard deviation encoder serves as a mapping from standard deviations across state dimensions to a low dimensional representation space. Each standard deviation $\sigma$ is computed across dimensions of the state $s\_{t}$. These deviations are then packed in a sequence $\{ \sigma\_{1}, \sigma\_{2}, ..., \sigma\_{T} \}$ and passed as inputs to the standard deviation encoder. Intuitively, the encoder learns changes across states in a batch of observations. This is similar to a dynamics model predicting future states, except that we map these states to a low dimensional embedding. We use a standard pyramid MLP network consisting of 2 hidden layers of 256 units each with ReLU non-linearity. Embeddings obtained from the encoder are concatenated with other latent representations and used by the final surprise encoder to estimate the surprise distribution.

---

### Official Review · Reviewer_rSgP · 2022-07-11

**Rating:** 5
**Confidence:** 2
**Soundness:** 3 good
**Presentation:** 3 good
**Contribution:** 3 good

**Summary:**

Surprise, which quantifies the degree of changes in agents' environment, has received significant attention in the case of SARL but remains an open problem for MARL. This paper explores surprise minimization in MARL by utilizing the free energy across all agents. The formulation of the energy-based model is theoretically akin to the minimum conjugate entropy objective. They further validate the theoretical claims in several MARL benchmarks.

**Questions:**

Some questions are listed above, here are other questions.

1. In Eq. (3), Why is there still a superscript $a$ after summing over $a$?

2. How to learn $V_{surp}^a(s,u,\sigma)$?  What is the definition and implementation of state deviation $\sigma$? What does the black and white grid in Figure 2 mean?

3. Why do SC2 and Predator-Prey need surprise minimization?

4. Can the authors illustrate more about "minimum conjugate entropy objective"?

**Limitations:**

The authors have described the limitations of this work. But in the view of reviewer, the main limitation is that some benchmarks are not suitable for surprise minimization. Maybe the reviewer does not well understand surprise minimization, hope the authors can give more intuitive explanations.

**Strengths And Weaknesses:**

Strengths

1. **Importance of the problem**. The problem this paper considered is important. How agents behave in the presence of sudden environmental changes is critical in both SARL and MARL. And this paper step forwards this topic in MARL.
2. **The novelty of the proposed method**. The proposed method is novel in a theoretical view. They have deeply investigated the influence of the introduction of energy-based surprise minimization in MARL.



Weaknesses

1. **Quality of empirical evaluation**. Not adequate. (1) The compared algorithms, QMIX, VDN, COMA, and IQL, are not SOTA anymore. More new baselines are expected, such as RODE, and QPLEX. (2) There are other difficult scenarios of SC2, what are the results of EMIX in these scenarios?
2. **Significance of results**. Not significant. In Table 1 and Table 2, EMIX only archives incremental improvements over outdated baselines.
3. **Clarity**. The writing need to be improved. For example, what does Figure 1 want to express? Why is the a superscript $a$ in $V_{surp}^a(s,u,\sigma)$, as there is no specific information related to agent $a$?

---

> ### Author Response · Authors · 2022-08-01
> **Response to Reviewer rSgP (2/2)**
>
> **How to learn $V^{a}\_{surp}(s,u,\sigma)$?**
> We use the quantity $E$ (line 13 in Algorithm 1) to jointly learn $V^{a}\_{surp}(s,u,\sigma)$ during MARL training. $E$ acts as intrinsic motivation as well as a learning signal to compute surprise. The estimate of surprise is computed using the architecture of Figure 2 where sampling from the surprise distribution is carried out using the reparameterization trick. Our framework learns $V^{a}\_{surp}(s,u,\sigma)$ much like any other value function (using a neural network), but by additionally undergoing the energy operator transformation $\mathcal{T} V^{a}\_{surp}(s,u,\sigma)$. This is achieved by using standard computation libraries. Since our code is implemented in PyTorch, we implement this as ```T_V = torch.logsumexp(V_surp, dim=1)```.
>
> **What is the definition and implementation of state deviation $\sigma$?**
> $\sigma$ denotes the standard deviations of observations for each agent $a$ (line 128). Implementation and computation of $\sigma$ can be found in Appendix D.1 (lines 623-626).
>
> **What does the black and white grid in Figure 2 mean?**
> Black and white grids represent low dimensional representations as 2D arrays (line 223, line 228). These are encoded embeddings obtained from encoders which are concatenated and used to estimate $V^{a}\_{surp}(s,u,\sigma)$.
>
> **Why do SC2 and Predator-Prey need surprise minimization?**
> In the case of SC2, micromanagement scenarios consist of a larger number of agents with different action spaces (line 254). Additionally, these scenarios present multiple opponents which introduce a greater degree of surprise within consecutive states (line 256). In the case of Predator Prey tasks, varying number of opponents helps us study surprise minimization under fixed dynamics (lines 318-320).
>
> **Can the authors illustrate more about "minimum conjugate entropy objective"?**
> The minimum conjugate entropy objective denotes the dual problem to surprise minimization. If we compute the Legendre Transform of our energy-based operator $\mathcal{T} V^{a}\_{surp}(s,u,\sigma) = \log \sum\_{a=1}^{N} \exp(V^{a}\_{surp}(s,u,\sigma))$ we obtain the entropy function $\mathcal{H}(x)$ where x is the gradient of the operator, $x = \mathcal{T} V^{a}\_{surp}(s,u,\sigma)$. This insight indicates that minimizing the energy operator $\mathcal{T} V^{a}\_{surp}(s,u,\sigma)$ is same as minimizing entropy in the space of gradients. Intuitively, our objective aims to minimize uncertainty in the learning signal. We have added the above discussion in Appendix E.5.
>
>
> Kindly let us know if our response above addresed your concerns.
>
> [1]. T. Wang, T. Gupta, A. Mahajan, B. Peng, S. Whiteson, C. Zhang, Rode: Learning roles to decompose multi-agent tasks, ICLR 2021.
>
> [2]. J. Wang, Z. Ren, T. Liu, Y. Yu, C. Zhang, Qplex: Duplex dueling multi-agent q-learning, ICLR 2021.
>
> [3]. C. Li, T. Wang, C. Wu, Q. Zhao, J. Yang, C. Zhang, Celebrating Diversity in Shared Multi-Agent Reinforcement Learning, NeurIPS 2021.
>
> [4]. K. Son, D. Kim, W. J. Kang, D. Hostallero, Y. Yi, QTRAN: Learning to Factorize with Transformation for Cooperative Multi-Agent Reinforcement learning, ICML 2019.
>
> [5]. Y. Wang, B. Han, T. Wang, H. Dong, C. Zhang, Dop: Off-policy multi-agent decomposed policy gradients, ICLR 2021.

---

> > ### Comment · Reviewer_rSgP · 2022-08-07
> > **Thanks for the detailed clarifications.**
> >
> > The responses have addressed most of my concerns. I'd like to raise the score.

---

> > > ### Author Response · Authors · 2022-08-07
> > > **Thank you and follow-up**
> > >
> > > Thank you! We would be happy to address your remaining concerns. Kindly let us know any changes/suggestions which would be required for you to support a stronger acceptance of our work.

---

> ### Author Response · Authors · 2022-08-01
> **Response to Reviewer rSgP (1/2)**
>
> We thank the respected reviewer for providing detailed feedback on the paper which is of utmost value to our work. We address your concerns below and in the general response comment above.
>
> **More new baselines are expected, such as RODE, and QPLEX.**
> Thank you for pointing this out. Yes, we agree that RODE and QPLEX are current state-of-the-art methods in MARL. However, we compare EMIX with traditional MARL baselines as these relate more closely to the problem of surprise minimization. Additionally, we aim to keep our comparison grounded towards end-to-end schemes and thus do not include baselines which have a design advantage over other methods. For instance, RODE employs a hierarchical scheme which pretrains action representations and role selectors. This gives RODE agents a significant advantage of pre-fixing roles rather than learning environment surprise end-to-end.
>
> Nevertheless, we believe RODE [1], QPLEX [2] and other CTDC methods [3,4,5] are important contributions in the MARL paradigm and merit discussion within this scope. To this end, we have added additional discussion in the form of related work in Appendix F. We aim to move this to the main text in the camera-ready version.
>
> **There are other difficult scenarios of SC2, what are the results of EMIX in these scenarios?**
> We restrict our analysis towards scenarios which present most surprise within a short temporal span. Additional SC2 scenarios either present longer temporal spans or more number of companion agents, both of which reduce the complexity and amount of surprise. Furthermore, these difficult scenarios focus on collaboration across agents rather than surprise minimization. An ideal scenario to evaluate surprise minimization would consist of a short duration, large number of opponents and rapidly changing states. This recipe is captured by our experiments on *so\_many\_baneling*, *5m vs 6m*, *8m vs 9m* and Predator-Prey tasks.
>
> **In Table 1 and Table 2, EMIX only archives incremental improvements over outdated baselines.**
> EMIX shows incremental yet consistent gains on SMAC benchmark over 5 random seeds. Note that we do not drop any seed. The reason behind marginal increments is that returns on SMAC tasks are maxing out. However, EMIX does present promising gains on more challenging *5m vs 6m*, *so many baneling* and *8m vs 9m* scenarios where the number of agents (and hence the surprise) is comparatively large. Additionally, the method shows good performance in particle world tasks. When compared to other surprise minimization schemes such SMiRL-QMIX, EMIX consistently presents higher returns. Furthermore, our qualitative results in Appendix E.1 depict that EMIX agents find surprise-robust policies when compared to QMIX.
>
> **what does Figure 1 want to express?**
> Figure 1 presents the intuition behind surprise minimization using the energy-based scheme. Interpreting the space of all surprising states as an energy landscape, MARL agents move from high energy states to low energy states which consist of minimum surprise. During training, agents train to find policies which not only provide rewarding actions, but also avoid risky states such as health damage or excessive enemy fire. Seeking these states leads to finding the minima on the energy landscape, as depicted in Figure 1. We have made our explanation of Figure 1 more intuitive.
>
> **Why is the a superscript $a$ in $V^{a}\_{surp}(s,u,\sigma)$, as there is no specific information related to agent $a$?**
> $V^{a}\_{surp}(s,u,\sigma)$ denotes the surprise value function which quantifies the surprise experienced by all agents (lines 130-132). We use $a$ as a superscript since $V^{a}\_{surp}(s,u,\sigma)$ fuses state and action information from all agents. This can be verified from its dependence on global states $s$, global actions $u$ and standard deviations $\sigma$. Figure 2 illustrates this dependence and its flow as well. Intuitively, $V^{a}\_{surp}(s,u,\sigma)$ is a $B \times N$ shaped array where $B$ is the batch size and $N$ is the number of agents. This array indicates the amount of surprise each agent experiences in a particular state.
>
> **In Eq. (3), Why is there still a superscript $a$ after summing over $a$?**
> $\mathcal{T} V^{a}\_{surp}(s,u,\sigma)$ denotes the application of energy operator $\mathcal{T} = \log \sum \exp$ on $V^{a}\_{surp}(s,u,\sigma)$. Since $\mathcal{T}$ is an operator, we use operator notation and retain the superscript $a$ on the operand. For instance, if $A$ is an operator on $x$, then we must write $Ax$.

---

> ### Author Response · Authors · 2022-08-05
> **Discussion**
>
> Kindly let us know if our response below addressed your concerns. We would be happy to discuss/update the work as per your comments.

---

### Author Response · Authors · 2022-08-01
**General Response to Reviewers**

We thank our respected reviewers for providing valuable feedback on the paper. The manuscript has been updated with the following changes highlighted in $\color{blue}{\text{blue}}$-

* Additional related work on CTDC MARL methods in Appendix F
* Discussion on Figure 1 in Section 4.1
* Discussion on minimum entropy conjugate objective in Appendix E.5
* Discussion on global state and standard deviation encoders in Appendix D.1
* Explanation of finite-horizon in Section 3
* Additional discussion and computation of Equation 17

---

### Meta-Review · Area_Chair_u5PK · 2022-09-09

**Recommendation:** Accept
**Confidence:** Certain

**Metareview:**

All reviewers appreciated the quality of the paper, its contributions, clarity and theoretical justification, and novelty. I agree with the reviewers in recommending acceptance.

**Award:**

No

---

### Decision · Program_Chairs · 2022-09-14

Accept